# Variationally Inferred Sampling through a Refined Bound

**DOI:** 10.3390/e23010123

**Published:** 2021-01-19

**Authors:** Víctor Gallego, David Ríos Insua

**Affiliations:** 1Institute of Mathematical Sciences (ICMAT), 28049 Madrid, Spain; david.rios@icmat.es; 2Statistical and Applied Mathematical Sciences Institute, Durham, NC 7333, USA; 3School of Management, University of Shanghai for Science and Technology, Shanghai 201206, China

**Keywords:** variational inference, MCMC, stochastic gradients, neural networks

## Abstract

In this work, a framework to boost the efficiency of Bayesian inference in probabilistic models is introduced by embedding a Markov chain sampler within a variational posterior approximation. We call this framework “refined variational approximation”. Its strengths are its ease of implementation and the automatic tuning of sampler parameters, leading to a faster mixing time through automatic differentiation. Several strategies to approximate evidence lower bound (ELBO) computation are also introduced. Its efficient performance is showcased experimentally using state-space models for time-series data, a variational encoder for density estimation and a conditional variational autoencoder as a deep Bayes classifier.

## 1. Introduction

Bayesian inference and prediction in large, complex models, such as in deep neural networks or stochastic processes, remains an elusive problem [1,2,3]. Variational approximations (e.g., automatic differentiation variational inference (ADVI) [4]) tend to be biased and underestimate uncertainty [5]. On the other hand, depending on the target distribution, Markov Chain Monte Carlo (MCMC) [6] methods, such as Hamiltonian Monte Carlo (HMC) [7]), tend to be exceedingly slow [8] in large scale settings with large amounts of data points and/or parameters. For this reason, in recent years, there has been increasing interest in developing more efficient posterior approximations [9,10,11] and inference techniques that aim to be as general and flexible as possible so that they can be easily used with any probabilistic model [12,13].

It is well known that the performance of a sampling method depends heavily on the parameterization used [14]. This work proposes a framework to automatically tune the parameters of a MCMC sampler with the aim of adapting the shape of the posterior, thus boosting the Bayesian inference efficiency. We deal with a case in which the latent variables or parameters are continuous. Our framework can also be regarded as a principled way to enhance the flexibility of variational posterior approximation in search of an optimally tuned MCMC sampler; thus the proposed name of our framework is the variationally inferred sampler (VIS).

The idea of preconditioning the posterior distribution to speed up the mixing time of a MCMC sampler has been explored recently in [15,16], where a parameterization was learned before sampling via HMC. Both papers extend seminal work in [17] by learning an efficient and expressive deep, non-linear transformation instead of a polynomial regression. However, they do not account for tuning the parameters of the sampler, as introduced in Section 3, where a fully, end-to-end differentiable sampling scheme is proposed.

The work presented in [18] introduced a general framework for constructing more flexible variational distributions, called normalizing flows. These transformations are one of the main techniques used to improve the flexibility of current variational inference (VI) approaches and have recently pervaded the approximate Bayesian inference literature with developments such as continuous-time normalizing flows [19] (which extend an initial simple variational posterior with a discretization of Langevin dynamics) or householder flow for mixtures of Gaussian distributions [20]. However, they require a generative adversarial network (GAN) [21] to learn the posterior, which can be unstable in high-dimensional spaces. We overcome this problem with our novel formulation; moreover, our framework is also compatible with different optimizers, rather than only those derived from Langevin dynamics [22]. Other recent proposals create more flexible variational posteriors based on implicit approaches typically requiring a GAN, as presented in [23] and including unbiased implicit variational inference (UIVI) [24] or semi-implicit variational inference (SIVI) [25]. Our variational approximation is also implicit but uses a sampling algorithm to drive the evolution of the density, combined with a Dirac delta approximation to derive an efficient variational approximation, as reported through extensive experiments in Section 5.

Closely related to our framework is the work presented in [26], where a variational autoencoder (VAE) is learned using HMC. We use a similar compound distribution as the variational approximation, yet our approach allows any stochastic gradient MCMC to be embedded, as well as facilitating the tuning of sampler parameters via gradient descent. Our work also relates to the recent idea of sampler amortization [27]. A common problem with these approaches is that they incur in an additional error—the amortization gap [28]—which we alleviate by evolving a set of particles through a stochastic process in the latent space after learning a good initial distribution, meaning that the initial approximation bias can be significantly reduced. A recent related article was presented in [29], which also defined a compound distribution. However, our focus is on efficient approximation using the reverse KL divergence, which allows sampler parameters to be tuned and achieves superior results. Apart from optimizing this kind of divergence, the main point is that we can compute the gradients of sampler parameters (Section 3.3), whereas in [29] the authors only consider a parameterless sampler: thus, our framework allows for greater flexibility, helping the user to tune sampler hyperparameters. In the Coupled Variational Bayes (CVB) [30] approach, optimization is in the dual space, whereas we optimize the standard evidence lower bound (ELBO). Note that even if the optimization was exact, the solutions would coincide, and it is not clear yet what happens in the truncated optimization case, other than performing empirical experiments on given datasets. We thus feel that there is room for implicit methods that perform optimization in the primal space (besides this, they are easier to implement). Moreover, the previous dual optimization approach requires the use of an additional neural network (see the paper on the Coupled Variational Bayes (CVB) approach or [31]). This adds a large number of parameters and requires another architecture decision. With VIS, we do not need to introduce an auxiliary network, since we perform a “non-parametric” approach by back-propagating instead through several iterations of SGLD. Moreover, the lack of an auxiliary network simplifies the design choices.

Thus, our contributions include a flexible and consistent variational approximation to the posterior, embedding an initial variational approximation within a stochastic process; an analysis of its key properties; the provision of several strategies for ELBO optimization using the previous approximation; and finally, an illustration of its power through relevant complex examples.

## 2. Background

Consider a probabilistic model p(x|z) and a prior distribution p(z), where *x* denotes the observations and $z\in R^{d}$ the unobserved latent variables or parameters, depending on the context. Whenever necessary for disambiguation purposes, we shall distinguish between *z* for latent variables and θ for parameters. Our interest is in performing inference regarding the unobserved *z* by approximating its posterior distribution
$$p\left(z|x\right)=\frac{p(z)p(x|z)}{\int p(z)p(x|z)dz}=\frac{p(x,z)}{p(x)}.$$

The integral assessing the evidence p(x)=∫p(z)p(x|z)dz is typically intractable. Thus, several techniques have been proposed to perform approximate posterior inference [3].

### 2.1. Inference as Optimization

Variational inference (VI) [4] tackles the problem of approximating the posterior p(z|x) with a tractable parameterized distribution $q_{\phi }\left(z|x\right)$. The goal is to find the parameters ϕ so that the variational distribution $q_{\phi }\left(z|x\right)$ (also referred to as variational guide or variational approximation) can be as close as possible to the actual posterior. Closeness is typically measured through the Kullback–Leibler divergence $KL(q_{\phi }||p)$, reformulated into the ELBO as follows:(1)$$\mathrm{ELBO}\left(q\right)=E_{q_{\phi }\left(z|x\right)}\left[\log p\left(x,z\right)-\log q_{\phi }\left(z|x\right)\right],$$
This is the objective to be optimized, typically through stochastic gradient descent techniques. To enhance flexibility, a standard choice for $q_{\phi }\left(z|x\right)$ is a Gaussian distribution $N(\mu_{\phi }\left(x\right),\sigma_{\phi }\left(x\right))$, with the mean and covariance matrix defined through a deep, non-linear model conditioned on observation *x*.

### 2.2. Inference as Sampling

HMC [7] is an effective sampling method for models whose probability is pointwise computable and differentiable. When scalability is an issue, as proposed by the authors in [32], a formulation of a continuous-time Markov process that converges to the target distribution p(z|x) can be used, which is based on the Euler–Maruyama discretization of Langevin dynamics
(2)$$\begin{array}{c} z_{t+1}\leftarrow z_{t}+\eta_{t}\nabla_{z}\log p\left(x,z_{t}\right)+N\left(0,2\eta_{t}I\right), \end{array}$$
where $\eta_{t}$ is the step size at time period *t*, and *I* is the identity matrix. The required gradient $\nabla \log p(z_{t},x)$ can be estimated using mini-batches of data. Several extensions of the original Langevin sampler have been proposed to increase its mixing speed, such as in [33,34,35,36]. We refer to these extensions as stochastic gradient MCMC samplers (SG-MCMC) [37].

## 3. A Variationally Inferred Sampling Framework

In standard VI, the variational approximation is analytically tractable and typically chosen as a factorized Gaussian, as mentioned above. However, it is important to note that other distributions can be adopted as long as they are easily sampled and their log-density and entropy values evaluated. However, in the rest of this paper, we focus on the Gaussian case, as the usual choice in the Bayesian deep learning community. Stemming from this variational approximation, we introduce several elements to construct the VIS.

Our first major modification of standard VI proposes the use of a more flexible distribution, approximating the posterior by embedding a sampler through
(3)$$q_{\phi ,\eta }\left(z|x\right)=\int Q_{\eta ,T}\left(z|z_{0}\right)q_{0,\phi }\left(z_{0}|x\right)dz_{0},$$
where $q_{0,\phi }\left(z|x\right)$ is the initial and tractable density $q_{\phi }\left(z|x\right)$ (i.e., the starting state for the sampler). We designate this as refined variational approximation. The conditional distribution $Q_{\eta ,T}\left(z|z_{0}\right)$ refers to a stochastic process parameterized by η and used to evolve the original density $q_{0,\phi }\left(z|x\right)$ for *T* periods, so as to achieve greater flexibility. Specific forms for $Q_{\eta ,T}\left(z|z_{0}\right)$ are described in Section 3.1. Observe that when T=0, no refinement steps are performed and the refined variational approximation coincides with the original one; on the other hand, as *T* increases, the approximation will be closer to the exact posterior, assuming that $Q_{\eta ,T}$ is a valid MCMC sampler in the sense of [37].

We next maximize a refined ELBO objective, replacing in Equation (Equation 1) the original $q_{\phi }$ by $q_{\phi ,\eta }$:(4)$$\mathrm{ELBO}\left(q_{\phi ,\eta }\right)=E_{q_{\phi ,\eta }\left(z|x\right)}\left[\log p\left(x,z\right)-\log q_{\phi ,\eta }\left(z|x\right)\right]$$
This is done to optimize the divergence $KL(q_{\phi ,\eta }\left(z|x\right)||p\left(z|x\right))$. The first term of Equation (Equation 4) requires only being able to sample from $q_{\phi ,\eta }\left(z|x\right)$; however, the second term, the entropy $-E_{q_{\phi ,\eta }\left(z|x\right)}\left[\log q_{\phi ,\eta }\left(z|x\right)\right]$, also requires the evaluation of the evolving, implicit density. Section 3.2 describes efficient methods to approximate this evaluation. As a consequence, performing variational inference with the refined variational approximation can be regarded as using the original variational guide while optimizing an alternative, tighter ELBO, as Section 4.2 shows.

The above facilitates a framework for learning the sampler parameters ϕ,η using gradient-based optimization, with the help of automatic differentiation [38]. For this, the approach operates in two phases. First, in a refinement phase, the sampler parameters are learned in an optimization loop that maximizes the ELBO with the new posterior. After several iterations, the second phase, focused on inference, starts. We allow the tuned sampler to run for sufficient iterations, as in SG-MCMC samplers. This is expressed algorithmically as follows.

Refinement phase:

Repeat the following until convergence:Sample an initial set of particles, $z_{0}∼q_{0,\phi }\left(z|x\right)$.Refine the particles through the sampler, $z_{T}∼Q_{\eta ,T}\left(z|z_{0}\right)$.Compute the ELBO objective from Equation (Equation 4).Perform automatic differentiation on the objective wrt parameters ϕ,η to update them.

Inference phase:

Once good sampler parameters $\phi^{*},\eta^{*}$ are learned,
Sample an initial set of particles, $z_{0}∼q_{0,\phi^{*}}\left(z|x\right)$.Use the MCMC sampler $z_{T}∼Q_{\eta^{*},T}\left(z|z_{0}\right)$ as T→∞.

Since the sampler can be run for a different number of steps depending on the phase, we use the following notation when necessary: VIS-*X*-*Y* denotes T=X iterations during the refining phase and T=Y iterations during the inference phase.

Let us specify now the key elements.

### 3.1. The Sampler $Q_{\eta ,T}\left(Z|Z_{0}\right)$


As the latent variables *z* are continuous, we evolve the original density $q_{0,\phi }\left(z|x\right)$ through a stochastic diffusion process [39]. To make it tractable, we discretize the Langevin dynamics using the Euler–Maruyama scheme, arriving at the stochastic gradient Langevin dynamics (SGLD) sampler (2). We then follow the process $Q_{\eta ,T}\left(z|z_{0}\right)$, which represents *T* iterations of the MCMC sampler.

As an example, for the SGLD sampler $z_{t}=z_{t-1}+\eta \nabla \log p\left(x,z_{t-1}\right)+\xi_{t},$ where *t* iterates from 1 to *T*. In this case, the only parameter is the learning rate η and the noise is $\xi_{t}∼N\left(0,2\eta I\right)$. The initial variational distribution $q_{0,\phi }\left(z|x\right)$ is a Gaussian parameterized by a deep neural network (NN). Then, after *T* iterations of the sampler *Q* are parameterized by η, we arrive at $q_{\phi ,\eta }$.

An alternative arises by ignoring the noise ξ [22], thus refining the initial variational approximation using only the stochastic gradient descent (SGD). Moreover, we can use Stein variational gradient descent (SVGD) [40] or a stochastic version [36] to apply repulsion between particles and promote more extensive explorations of the latent space.

### 3.2. Approximating the Entropy Term

We propose four approaches for the ELBO optimization which take structural advantage of the refined variational approximation.

#### 3.2.1. Particle Approximation (VIS-P)

In this approach, we approximate the posterior $q_{\phi ,\eta }\left(z|x\right)$ by a mixture of Dirac deltas (i.e., we approximate it with a finite set of particles), by sampling $z^{\left(1\right)},\ldots ,z^{\left(M\right)}∼q_{\phi ,\eta }\left(z|x\right)$ and setting
$$q_{\phi ,\eta }\left(z|x\right)=\frac{1}{M}\sum_{m=1}^{M}\delta \left(z-z^{\left(m\right)}\right).$$

In this approximation, the entropy term in (4) is set to zero. Consequently, the sample converges to the maximum posterior (MAP). This may be undesirable when training generative models, as the generated samples usually have little diversity. Thus, in subsequent computations, we add to the refined ELBO the entropy of the initial variational approximation, $E_{q_{0,\phi }\left(z|x\right)}\left[\log q_{0,\phi }\left(z|x\right)\right]$, which serves as a regularizer alleviating the previous problem. When using SGD as the sampler, the resulting ELBO is tighter than that without refinement, as shown in Section 4.2.

#### 3.2.2. MC Approximation (VIS-MC)

Instead of performing the full marginalization in Equation (Equation 3), we approximate it with $q_{\phi ,\eta }\left(z_{T},\ldots ,z_{0}|x\right)=\prod_{t=1}^{T}q_{\eta }\left(z_{t}|z_{t-1}\right)q_{0,\phi }\left(z_{0}|x\right)$; i.e., we consider the joint distribution for the refinement. However, in inference we only keep the $z_{T}$ values. The entropy for each factor in this approximation is straightforward to compute. For example, for the SGLD case, we have
$$z_{t}=z_{t-1}+\eta \nabla \log p\left(x,z_{t-1}\right)+N\left(0,2\eta I\right),\;t=1,\ldots ,T.$$
This approximation tracks a better estimate of the entropy than VIS-P, as we are not completely discarding it; rather, for each *t*, we marginalize out the corresponding $z_{t}$ using one sample.

#### 3.2.3. Gaussian Approximation (VIS-G)

This approach is targeted at settings in which it could be helpful to have a posterior approximation that places density over the whole *z* space. In the specific case of using SGD as the inner kernel, we have
$$\begin{array}{cc} z_{0} & ∼q_{0,\phi }\left(z_{0}|x\right)=N\left(z_{0}|\mu_{\phi }\left(x\right),\sigma_{\phi }\left(x\right)\right)\\ z_{t} & =z_{t-1}+\eta \nabla \log p\left(x,z_{t-1}\right),\;t=1,\ldots ,T. \end{array}$$

By treating the gradient terms as points, the refined variational approximation can be computed as $q_{\phi ,\eta }\left(z|x\right)=N\left(z|z_{T},\sigma_{\phi }\left(x\right)\right)$. Observe that there is an implicit dependence on η through $z_{T}$.

#### 3.2.4. Fokker–Planck Approximation (VIS-FP)

Using the Fokker–Planck equation, we derive a deterministic sampler via iterations of the form
$$z_{t}=z_{t-1}+\eta \left(\nabla \log p\left(x,z_{t-1}\right)-\nabla \log q_{t}\left(z_{t-1}\right)\right),\;t=1,\ldots ,T.$$

Then, we approximate the density $q_{\phi ,\eta }\left(z|x\right)$ using a mixture of Dirac deltas. A detailed derivation of this approximation is given in Appendix A.

### 3.3. Back-Propagating through the Sampler

In standard VI, the variational approximation q(z|x;ϕ) is parameterized by ϕ. The parameters are learned employing SGD, or variants such as Adam [41], using the gradient $\nabla_{\phi }\mathrm{ELBO}\left(q\right)$. We have shown how to embed a sampler inside the variational guide. It is therefore also possible to compute a gradient of the objective with respect to the sampler parameters η (see Section 3.1). For instance, we can compute a gradient $\nabla_{\eta }\mathrm{ELBO}\left(q\right)$ with respect to the learning rate η from the SGLD or SGD processes to search for an optimal step size at every VI iteration. This is an additional step apart from using the gradient $\nabla_{\phi }\mathrm{ELBO}\left(q\right)$ which is used to learn a good initial sampling distribution.

## 4. Analysis of Vis

Below, we highlight key properties of the proposed framework.

### 4.1. Consistency

The VIS framework is geared towards SG-MCMC samplers, where we can compute the gradients of sampler hyperparameters to speed up mixing time (a common major drawback in MCMC [42]). After back-propagating for a few iterations through the SG-MCMC sampler and learning a good initial distribution, one can resort to the learned sampler in the second phase, so standard consistency results from SG-MCMC apply as T→∞ [43].

### 4.2. Refinement of ELBO

Note that, for a refined guide using the VIS-P approximation and M=1 samples, the refined objective function can be written as
$$E_{q(z_{0}|x)}\left[\log p\left(x,z_{0}+\eta \nabla \log p\left(x,z_{0}\right)\right)-\log q\left(z_{0}|x\right)\right]$$
noting that $z=z_{0}+\eta \nabla \log p\left(x,z_{0}\right)$ when using SGD for T=1 iterations. This is equivalent to the refined ELBO in (Equation 4). Since we are perturbing the latent variables in the steepest direction, we show easily that, for a moderate η, the previous bound is tighter than $E_{q(z_{0}|x)}\left[\log p\left(x,z_{0}\right)-\log q\left(z_{0}|x\right)\right]$, the one for the original variational guide $q(z_{0}|x)$. This reformulation of ELBO is also convenient since it provides a clear way of implementing our refined variational inference framework in any probabilistic programming language (PPL) supporting algorithmic differentiation.

Respectively, for the VIS-FP case, we find that its deterministic flow follows the same trajectories as SGLD: based on standard results of MCMC samplers [44], we have
$$KL(q_{\phi ,\eta }\left(z|x\right)||p\left(z|x\right))\leq KL(q_{0,\phi }\left(z|x\right)||p\left(z|x\right)).$$

A similar reasoning applies to the VIS-MC approximation; however, it does not hold for VIS-G since it assumes that the posterior is Gaussian.

### 4.3. Taylor Expansion

This analysis applies only to VIS-P and VIS-FP. As stated in Section 4.2, within the VIS framework, optimizing the ELBO resorts to the performance of $\max_{z}\log p\left(x,z+\Delta z\right)$, where Δz is one iteration of the sampler; i.e., Δz=η∇logp(x,z) in the SGD case (VIS-P), or Δz=η∇(logp(x,z)−logq(z)) in the VIS-FP case. For notational clarity, we consider the case T=1, although a similar analysis follows in a straightforward manner if more refinement steps are performed.

Consider a first-order Taylor expansion of the refined objective
$$\log p\left(x,z+\Delta z\right)\approx \log p\left(x,z\right)+{\left(\Delta z\right)}^{⊺}\nabla \log p\left(x,z\right).$$

Taking gradients with respect to the latent variables *z*, we arrive at
$$\nabla_{z}\log p\left(x,z+\Delta z\right)\approx \nabla_{z}\log p\left(x,z\right)+\eta \nabla_{z}\log p{\left(x,z\right)}^{⊺}\nabla_{z}^{2}\log p\left(x,z\right),$$
where we have not computed the gradient through the Δz term (i.e., we treated it as a constant for simplification). Then, the refined gradient can be deemed to be the original gradient plus a second order correction. Instead of being modulated by a constant learning rate, this correction is adapted by the chosen sampler. The experiments in Section 5.4 show that this is beneficial for the optimization as it typically takes fewer iterations than the original variant to achieve lower losses.

By further taking gradients through the Δz term, we may tune the sampler parameters such as the learning rate as presented in Section 3.3. Consequently, the next subsection describes two differentiation modes.

### 4.4. Two Automatic Differentiation Modes for Refined ELBO Optimization

For the first variant, remember that the original variant can be rewritten (which we term Full AD) as
(5)$$E_{q}\left[\log p(x,z+\Delta z)-\log q(z+\Delta z|x)\right].$$

We now define a stop gradient operator ⊥ (which corresponds to detach in Pytorch or stop_gradient in tensorflow) that sets the gradient of its operand to zero—i.e., $\nabla_{x}⊥\left(x\right)=0$—whereas in a forward pass, it acts as the identity function—that is, ⊥(x)=x. With this, a variant of the ELBO objective (which we term Fast AD) is
(6)$$E_{q}\left[\log p(x,z+⊥(\Delta z))-\log q(z+⊥(\Delta z)|x)\right].$$

Full AD ELBO enables a gradient to be computed with respect to the sampler parameters inside Δz at the cost of a slight increase in computational burden. On the other hand, the Fast AD variant may be useful in numerous scenarios, as illustrated in the experiments.

#### Complexity

Since we need to back propagate through *T* iterations of an SG-MCMC scheme, using standard results of meta-learning and automatic differentiation [45], the time complexity of our more intensive approach (Full-AD) is O(mT), where *m* is the dimension of the hyperparameters (the learning rate of SG-MCMC and the latent dimension). Since for most use cases, the hyperparameters lie in a low-dimensional space, the approach is therefore scalable.

## 5. Experiments

The following experiments showcase the power of our approach as well as illustrating the the impact of various parameters on its performance, guiding their choice in practice. We also present a comparison with standard VIS and other recent variants, showing that the increased computational complexity of computing gradients through sampling steps is worth the gains in flexibility. Moreover, the proposed framework is compatible with other structured inference techniques, such as the sum–product algorithm, as well as serving to support other tasks such as classification.

Within the spirit of reproducible research, the code for VIS has been released at https://github.com/vicgalle/vis. The VIS framework is implemented with Pytorch [46], although we have also released a notebook for the first experiment using Jax to highlight the simple implementation of VIS. In any case, we emphasize that the approach facilitates rapid iterations over a large class of models.

### 5.1. Funnel Density

We first tested the framework on a synthetic yet complex target distribution. This experiment assessed whether VIS is suitable for modeling complex distributions. The target bi-dimensional density was defined through
$$\begin{array}{cc} z_{1} & ∼N(0,1.35)\\ z_{2} & ∼N(0,\exp\left(z_{1}\right)). \end{array}$$

We adopted the usual diagonal Gaussian distribution as the variational approximation. For VIS, we used the VIS-P approximation and refined it for T=1 steps using SGLD. Figure 1 top shows the trajectories of the lower bound for up to 50 iterations of variational optimization with Adam: our refined version achieved a tighter bound. The bottom figures present contour curves of the learned variational approximations. Observe that the VIS variant was placed closer to the mean of the true distribution and was more disperse than the original variational approximation, illustrating the fact that the refinement step helps in attaining more flexible posterior approximations.

### 5.2. State-Space Markov Models

We tested our variational approximation on two state-space models: one for discrete data and another for continuous observations. These experiments also demonstrated that the framework is compatible with standard inference techniques such as the sum–product scheme from the Baum–Welch algorithm or Kalman filter. In both models, we performed inference on their parameters θ. All the experiments in this subsection used the Fast AD version (Section 4.4) as it was not necessary to further tune the sampler parameters to obtain competitive results. Full model implementations can be found in Section B.1, based on funsor (https://github.com/pyro-ppl/funsor/), a PPL on top of the Pytorch autodiff framework.

Hidden Markov Model (HMM): The model equations are
(7)$$p\left(x_{1:\tau },z_{1:\tau },\theta \right)=\prod_{t=1}^{\tau }p\left(x_{t}|z_{t},\theta_{em}\right)p\left(z_{t}|z_{t-1},\theta_{tr}\right)p\left(\theta \right),$$
where each conditional is a categorical distribution taking five different classes. The prior is $p\left(\theta \right)=p\left(\theta_{em}\right)p\left(\theta_{tr}\right)$ based on two Dirichlet distributions that sample the observation and state transition probabilities, respectively.

Dynamic Linear Model (DLM): The model equations are as in (Equation 7), although the conditional distributions are now Gaussian and the parameters θ refer to the observation and transition variances.

For each model, we generated a synthetic dataset and used the refined variational approximation with T=0,1,2. For the original variational approximation to the parameters θ, we used a Dirac delta. Performing VI with this approximation corresponded to MAP estimation using the Baum–Welch algorithm in the HMM case [47] and the Kalman filter in the DLM case [48], as we marginalized out the latent variables $z_{1:\tau }$. We used the VIS-P variant since it was sufficient to show performance gains in this case.

Figure 2 shows the results. The first row reports the experiments related to the HMM, the second row those for the DLM. We report the evolution of the log-likelihood during inference in all graphs; the first column reports the number of ELBO iterations, and the second column portrays clock times as the optimization takes place. They confirm that VIS (T>0) achieved better results than standard VI (T=0) for a comparable amount of time. Note also that there was not as much gain when changing from T=1 to T=2 as there is from T=0 to T=1, suggesting the need to carefully monitor this parameter. Finally, the top-right graph for the case T=0 is shorter as it requires less clock time.

#### 5.2.1. Prediction with an HMM

With the aim of assessing whether ELBO optimization helps in attaining better auxiliary scores, results in a prediction task are also reported. We generated a synthetic time series of alternating values of 0 and 1 for τ=105 timesteps. We trained the previous HMM model on the first 100 points and report in Table 1 the accuracy of the predictive distribution $p(y_{t})$ averaged over the final five time-steps. We also report the predictive entropy as it helps in assessing the confidence of the model in its predictions, as a strictly proper scoring rule [49]. To guarantee the same computational budget time and a fair comparison, the model without refinement was run for 50 epochs (an epoch was a full iteration over the training dataset), whereas the model with refinement was run for 20 epochs. It can be observed that the refined model achieved higher accuracy than its counterpart. In addition, it was more correctly confident in its predictions.

#### 5.2.2. Prediction with a DLM

We tested the VIS framework on Mauna Loa monthly CO${}_{2}$ time-series data [50]. We used the first 10 years as a training set, and we tested over the next 2 years. We used a DLM composed of a local linear trend plus a seasonal block of periodicity 12. Data were standardized to a mean of zero and standard deviation of one. To guarantee the same computational budget time, the model without refining was run for 10 epochs, whereas the model with refinement was run for 4 epochs. Table 2 reports the mean absolute error (MAE) and predictive entropy. In addition, we computed the interval score [49], as a strictly proper scoring rule. As can be seen, for similar clock times, the refined model not only achieved a lower MAE, but also its predictive intervals were narrower than the non-refined counterpart.

### 5.3. Variational Autoencoder

The third batch of experiments showed that VIS was competitive with respect to other algorithms from the recent literature, including unbiased implicit variational inference (UIVI [24]), semi-implicit variational inference (SIVI [25]), variational contrastive divergence (VCD [29]), and the HMC variant from [26], showing that our framework can outperform those approaches in similar experimental settings.

To this end, we tested the approach with a variational autoencoder (VAE) model [51]. The VAE defines a conditional distribution $p_{\theta }\left(x|z\right)$, generating an observation *x* from a latent variable *z* using parameters θ. For this task, our interest was in modeling the 28×28 image distributions underlying the MNIST [52] and the fashion-MNIST [53] datasets. To perform inference (i.e., to learn the parameters θ) the VAE introduces a variational approximation $q_{\phi }\left(z|x\right)$. In the standard setting, this distribution is Gaussian; we instead used the refined variational approximation comparing various values of *T*. We used the VIS-MC approximation (although we achieved similar results with VIS-G) with the Full AD variant given in Section 4.4.

For the experimental setup, we reproduced the setting in [24]. For $p_{\theta }\left(x|z\right)$, we used a factorized Bernoulli distribution parameterized by a two layer feed-forward network with 200 units in each layer and relu activations, except for a final sigmoid activation. As a variational approximation $q_{\phi }\left(z|x\right)$, we used a Gaussian with mean and (diagonal) covariance matrix parameterized by two distinct neural networks with the same structure as previously used, except for sigmoid activation for the mean and a softplus activation for the covariance matrix.

Results are reported in Table 3. To guarantee fair comparison, we trained the VIS-5-10 variant for 10 epochs, whereas all the other variants were trained for 15 epochs (fMNIST) or 20 epochs (MNIST), so that the VAE’s performance was comparable to that reported in [24]. Although VIS was trained for fewer epochs, by increasing the number *T* of MCMC iterations, we dramatically improved the test log-likelihood. In terms of computational complexity, the average time per epoch using T=5 was 10.46 s, whereas with no refinement (T=0), the time was 6.10 s (which was the reason behind our decision to train the refined variant for fewer epochs): a moderate increase in computing time may be worth the dramatic increase in log-likelihood while not introducing new parameters into the model, except for the learning rate η.

Finally, as a visual inspection of the VAE reconstruction quality trained with VIS, Figure 3 and Figure 4, respectively, display 10 random samples of each dataset.

### 5.4. Variational Autoencoder as a Deep Bayes Classifier

In the final experiments, we investigated whether VIS can deal with more general probabilistic graphical models and also perform well in other inference tasks such as classification. We explored the flexibility of the proposed scheme to solve inference problems in an experiment with a classification task in a high-dimensional setting with the MNIST dataset. More concretely, we extended the VAE model, conditioning it on a discrete variable y∈Y={0,1,…,9}, leading to a conditional VAE (cVAE). The cVAE defined a decoder distribution $p_{\theta }\left(x|z,y\right)$ on an input space $x\in R^{D}$ given a class label y∈Y, latent variables $z\in R^{d}$ and parameters θ. Figure 5 depicts the corresponding probabilistic graphic model. Additional details regarding the model architecture and hyperparameters are given in Appendix B.

To perform inference, a variational posterior was learned as an encoder $q_{\phi }\left(z|x,y\right)$ from a prior p(z)∼N(0,I). Leveraging the conditional structure on *y*, we used the generative model as a classifier using the Bayes rule,
(8)$$p\left(y|x\right)\propto p\left(y\right)p\left(x|y\right)=p\left(y\right)\int p_{\theta }\left(x|z,y\right)q_{\phi }\left(z|x,y\right)dz\approx \frac{1}{M}\sum_{m=1}^{M}p_{\theta }\left(x|z^{\left(m\right)},y\right)p\left(y\right),$$
where we used *M* Monte Carlo samples $z^{\left(m\right)}∼q_{\phi }\left(z|x,y\right)$. In the experiments, we set M=5. Given a test sample *x*, the label $\hat{y}$ with the highest probability p(y|x) is predicted.

For comparison, we performed several experiments changing *T* in the transition distribution $Q_{\eta ,T}$ of the refined variational approximation. The results are given in Table 4, which reports the test accuracy at end of the refinement phase. Note that we are comparing different values of *T* depending on their use in refinement or inference phases (in the latter, the model and variational parameters were kept frozen). The model with $T_{ref}=5$ was trained for 10 epochs, whereas the other settings were for 15 epochs, to give all settings a similar training time. Results were averaged over three runs with different random seeds. In all settings, we used the VIS-MC approximation for the entropy term. From the results, it is clear that the effect of using the refined variational approximation (the cases when T>0) is crucially beneficial to achieve higher accuracy. The effect of learning a good initial distribution and inner learning rate by using the gradients $\nabla_{\phi }\mathrm{ELBO}\left(q\right)$ and $\nabla_{\eta }\mathrm{ELBO}\left(q\right)$ has a highly positive impact in the accuracy obtained.

On a final note, we have not included the case of only using an SGD or an SGLD sampler (i.e., without learning an initial distribution $q_{0,\phi }\left(z|x\right)$) since the results were much worse than those in Table 4 for a comparable computational budget. This strongly suggests that, for inference in high-dimensional, continuous latent spaces, learning a good initial distribution through VIS may accelerate mixing time dramatically.

## 6. Conclusions

In this work, we have proposed a flexible and efficient framework to perform large-scale Bayesian inference in probabilistic models. The scheme benefits from useful properties and can be employed to efficiently perform inference with a wide class of models such as state-space time series, variational autoencoders and variants such as the conditioned VAE for classification tasks, defined through continuous, high-dimensional distributions.

The framework can be seen as a general approach to tuning MCMC sampler parameters, adapting the initial distributions and learning rate. Key to the success and applicability of the VIS framework are the ELBO approximations based on the introduced refined variational approximation, which are computationally cheap but convenient.

Better estimates of the refined density and its gradient may be a fruitful line of research, such as the spectral estimator used in [54]. Another alternative is to use a deterministic flow (such as SGD or SVGD), keeping track of the change in entropy at each iteration using the change of the variable formula, as in [55]. However, this requires a costly Jacobian computation, making it unfeasible to combine with our approach of back-propagation through the sampler (Section 3.3) for moderately complex problems. We leave this for future exploration. Another interesting and useful line of further research would be to tackle the case in which the latent variables *z* are discrete. This would entail adapting the automatic differentiation techniques to be able to back-propagate the gradients through the sequences of acceptance steps necessary in Metropolis–Hastings samplers.

In order to deal with the implicit variational density, it may be worthwhile to consider optimizing the Fenchel dual of the KL divergence, as in [31]. However, this requires the use of an auxiliary neural network, which may entail a large computational price compared with our simpler particle approximation.

Lastly, probabilistic programming offers powerful tools for Bayesian modeling. A PPL can be viewed as a programming language extended with random sampling and Bayesian conditioning capabilities, complemented with an inference engine that produces answers to inference, prediction and decision-making queries. Examples include WinBUGS [56], Stan [57] or the recent Edward [58] and Pyro [59] languages. We plan to adapt VIS into several PPLs to facilitate the adoption of the framework.

## Figures and Tables

**Figure 1 entropy-23-00123-f001:**
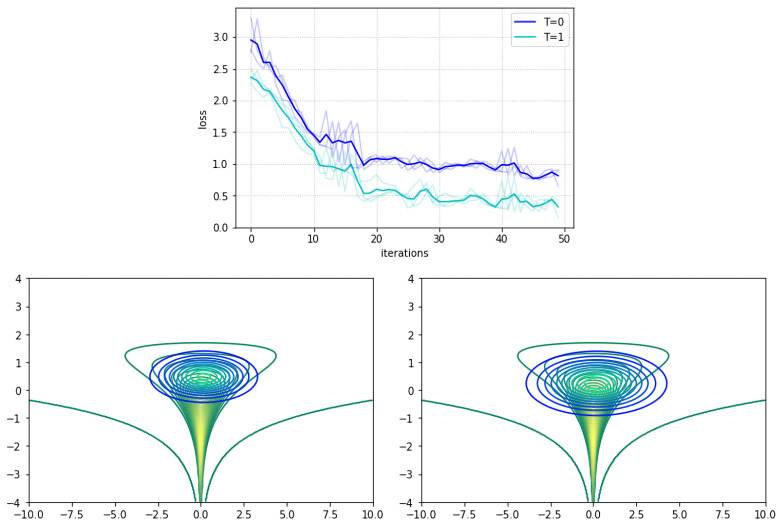
**Top**: Evolution of the negative evidence lower bound (ELBO) loss objective over 50 iterations. Darker lines depict means along different seeds (lighter lines). **Bottom left**: Contour curves (blue–turquoise) of the variational approximation with no refinement (T=0) at iteration 30 (loss of 1.011). **Bottom right**: Contour curves (blue–turquoise) of refined variational approximation (T=1) at iteration 30 (loss of 0.667). Green–yellow curves denote target density.

**Figure 2 entropy-23-00123-f002:**
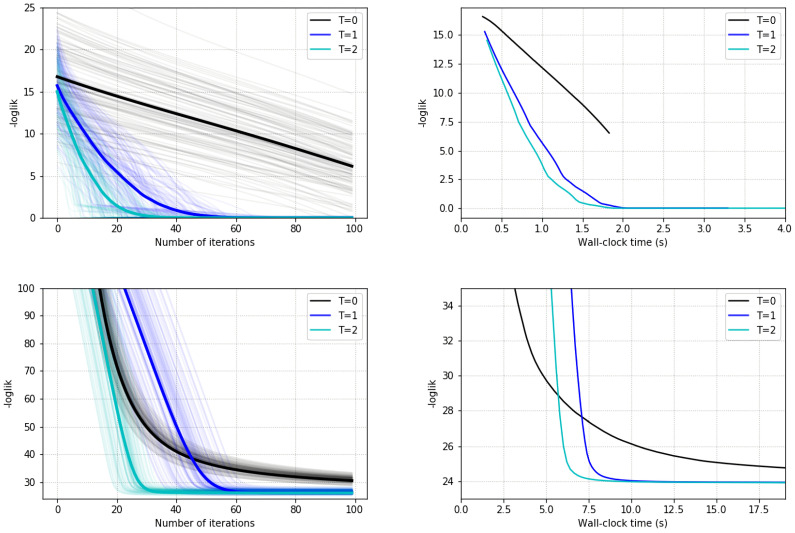
Results of ELBO optimization for state-space models. **Top-left** (Hidden Markov Model (HMM)): Log-likelihood against the number of ELBO gradient iterations. **Top-right** (HMM): Log-likelihood against clock time. **Bottom-left** (Dynamic Linear Model (DLM)): Log-likelihood against number of ELBO gradient iterations. **Bottom-right** (DLM): Log-likelihood against against clock time.

**Figure 3 entropy-23-00123-f003:**
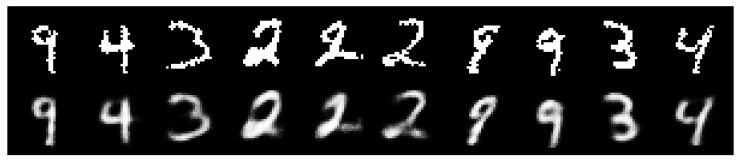
Top: original images from MNIST. Bottom: reconstructed images using VIS-5-10 at 10 epochs.

**Figure 4 entropy-23-00123-f004:**
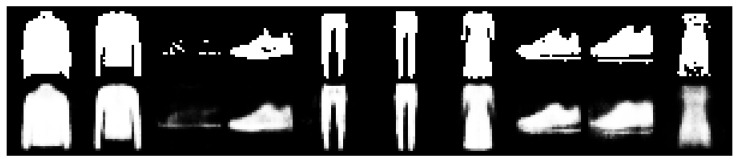
Top: original images from fMNIST. Bottom: reconstructed images using VIS-5-10 at 10 epochs.

**Figure 5 entropy-23-00123-f005:**
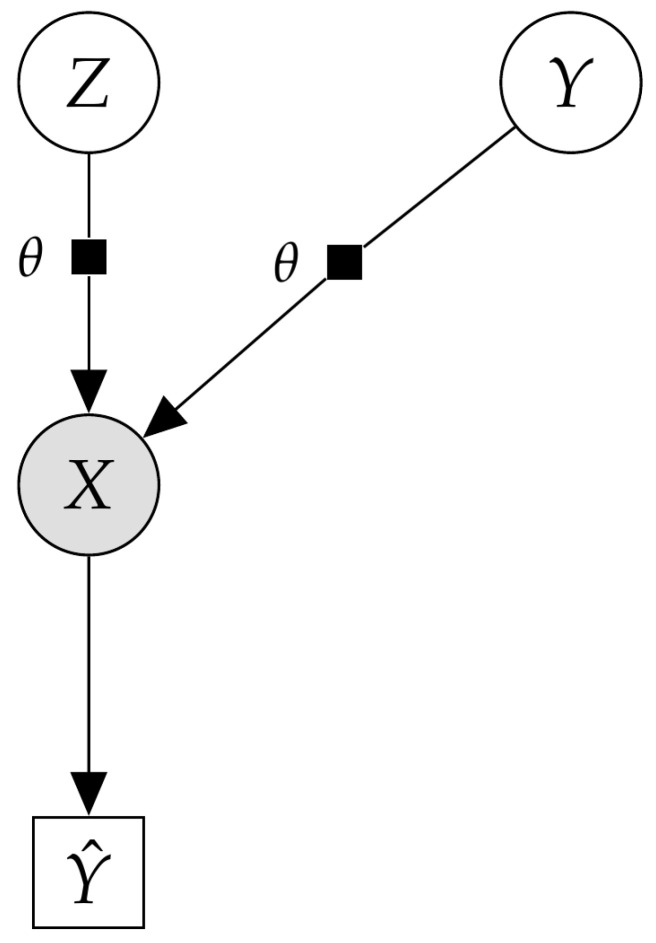
Probabilistic graphical model for the deep Bayes classifier.

**Table 1 entropy-23-00123-t001:** Prediction metrics for the HMM.

	T=0	T=1
accuracy	0.40	0.84
predictive entropy	1.414	1.056
logarithmic score	−1.044	−0.682

**Table 2 entropy-23-00123-t002:** Prediction metrics for the DLM.

	T=0	T=1
MAE	0.270	0.239
predictive entropy	2.537	2.401
interval score (α=0.05)	15.247	13.461

**Table 3 entropy-23-00123-t003:** Test log-likelihood on binarized MNIST and fMNIST. Bold numbers indicate the best results. UIVI: unbiased implicit variational inference; SIVI: semi-implicit variational inference; VAE: variational autoencoder; VCD: variational contrastive divergence; HMC-DLGM: Hamiltonian Monte Carlo for Deep Latent Gaussian Models; VIS: variationally inferred sampler.

Method	MNIST	fMNIST
Results from [24]		
UIVI	−94.09	−110.72
SIVI	−97.77	−121.53
VAE	−98.29	−126.73
Results from [29]		
VCD	−95.86	−117.65
HMC-DLGM	−96.23	−117.74
This paper		
VIS-5-10	−82.74±0.19	−105.08±0.34
VIS-0-10	−96.16±0.17	−120.53±0.59
VAE (VIS-0-0)	−100.91±0.16	−125.57±0.63

**Table 4 entropy-23-00123-t004:** Results on digit classification task using a deep Bayes classifier.

$T_{\mathrm{ref}}$	$T_{\inf}$	Acc. (Test)
0	0	96.5±0.5 %
0	10	97.7±0.7 %
5	10	99.8±0.2 %

## Data Availability

Not applicable.

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
