# Peer review of "Variationally Inferred Sampling through a Refined Bound"

_entropy, 2021, doi:10.3390/e23010123_

Round 1

Reviewer 1 Report

This paper discusses another trick for improving VI; the main idea is to evolve the variational posterior using a MCMC-type sampler. The paper is mostly well-written and appears technically sound. The paper has previously been presented at a workshop, and at a few places I get the feeling that the strict page limitations typically enforced at such events results in the current version also being a bit dense -> Some additional text would be welcome. Nevertheless, I find the paper interesting and publishable aften the fairly simple updates I discuss below; I use line numbers as presented in the PDF even though they don't really count every line:

  • Eq 2: Spell out that the gradient is wrt z. 
  • Line 47: Here and also other places it seems you consider q_0 to be a Gaussian as "The Choice". While a Gaussian is quite popular, I don't think you need this assumption, and propose you remove these statements.
  • Line 55: "Its first term" --> "The first term of Equation (4)"
  • Line 80: What happens with discrete Z? I believe it would be OK if your setup only works for continuous Z, but it is not clear, and should be discussed.
  • Line 97: "... the entropy term in (4) is zero...". Is there a proof of that, or a cite you can add? I can't help thinking that the integral (seen as the limit of  a Riemann sum) will diverge. Please prove the claim, give a specific cite, or (if I am right) discuss what goes on in VIS-P when the entropy is unbounded.
  • Line 103: "... we approximate it through...": How does this work? The "intermediate variables" z_1, ... z_{T-1} must be marginalized out. Maybe show how this works for the SGLD in a worked example. 
  • Line 109: Since the VIS-FP setup ends with a mixture of Dirac deltas (as VIS-P) I assume we have the same issues with calculating the entropy here as you did with VIS-P? What is your solution now?
  • Line 233: The generative model for the HMM appears to be wrong. p(x_t|x_{t-1}) should probably rather relate to z
  • Line 310: I found the introduction of influence diagrams to be rather far-fetched. Your model is simply a classifier with the classification rule that we allocate to the most probable class. It can be the solution to an influence diagram, but an influence diagram is (typically) much richer than what you use here; we for instance don't even get the utility-function defined, and the purpose of \hat{Y} in the model is unclear in the text. I propose to drop the Influence Diagram claim, and simply relate to this as what it is: A classifier. 

Reviewer 2 Report

See attached file
